# OpenReview forum: "Distribution-Aware Robust Bilevel Optimization: Quantile-Guided Huber Updates in Two-Timescale Stochastic Approximation"
_ICML.cc/2026/Conference — Submitted to ICML 2026_

### Official Review · Reviewer_L3Hr · 2026-03-11

**Soundness:** 3
**Presentation:** 3
**Significance:** 3
**Originality:** 3
**Overall Recommendation:** 4
**Confidence:** 3

**Summary:**

The paper tackles the significant challenge of instability in stochastic bilevel optimization (BLO) when subjected to heavy-tailed, infinite-variance noise. Standard Two-Timescale Stochastic Approximation (TTSA) assumes bounded variance , and existing adaptive methods often fail to distinguish between informative geometric signals and stochastic outliers. To address this, the authors introduce Robust Quantile-guided TTSA (RQ-TTSA). This method utilizes a history buffer of gradient norms to dynamically estimate a quantile threshold, applying a norm-based, Huber-style clipping operator to bound the effective variance while preserving the optimization trajectory.

Theoretically, the authors establish a global convergence rate of $\mathcal{O}(T^{-\frac{p-1}{3p-2}})$ for non-convex strongly-convex objectives under heavy-tailed noise ($p \in (1,2]$), matching the optimal dependence. Empirically, RQ-TTSA demonstrates superior stability and performance across six varied tasks—including synthetic landscapes, natural label shift, momentum poisoning in games, and offline reinforcement learning—while incurring a negligible computational overhead of approximately 2.7%.

**Compliance With Llm Reviewing Policy:**

Affirmed.

**Final Justification:**

I appreciate the authors' response. My questions have been well addressed, and I will maintain my original positive score.

**Key Questions For Authors:**

1. Algorithm 1 uses a rolling empirical quantile threshold computed from a finite history buffer, whereas the main theory assumes a prescribed threshold growth rule. Can the authors provide a theorem that directly covers the implemented adaptive quantile estimator, or clearly state the assumptions needed to justify the current analysis for the actual algorithm?
2. In Appendix A.5, why does clipping the lower-level stochastic gradient imply an almost-sure bound on the hypergradient estimator $\||\hat{\nabla}\Phi_k\||$? This step seems crucial for the Bernstein-style high-probability argument.
3. In Theorem A.5, the proof appears to depend on $E\||g\||^p$, including a term involving $\||v\||^p$, while the theorem statement is written only in terms of $\sigma^p$ and $\psi$. Is the theorem statement missing dependence on the gradient magnitude, or is an additional assumption required?
4. Please clarify the hyperparameter tuning protocol across baselines. Were all methods given the same search budget and selection criterion on each benchmark?
5. Can the authors provide direct empirical evidence that the non-synthetic benchmarks indeed exhibit heavy-tailed gradient behavior, for example through gradient-norm distributions or tail-index diagnostics?

**Limitations:**

yes

**Strengths And Weaknesses:**

## Strengths
- The article studies an important concept: robustness of stochastic bilevel optimization under heavy-tailed noise, which is practically relevant in settings such as reinforcement learning and heterogeneous learning where bounded-variance assumptions may be unrealistic.
- The paper's principal contribution pertains to introducing a simple distribution-aware clipping mechanism for TTSA: a rolling history buffer is used to estimate a quantile-based threshold, and the lower-level stochastic gradient is clipped via a Huber-style operator. The idea is intuitive, easy to implement, and appears potentially usable as a plug-in component in other first-order or momentum-based bilevel methods.
- The empirical evaluation is relatively broad. Beyond synthetic heavy-tail stress tests, the paper reports results on USPS, Fashion-MNIST, a zero-sum game with momentum poisoning, and offline actor-critic optimization on LunarLander, and also includes sensitivity and runtime analyses. This breadth is a positive aspect of the submission.
- Several experimental results are promising. In particular, the method performs strongly on Fashion-MNIST, improves robustness in the momentum-poisoning setup, and is reported to incur only modest overhead from maintaining the history buffer.

## Weaknesses
- A main concern is a theory-to-algorithm mismatch. Algorithm 1 uses a rolling empirical quantile computed from a finite history buffer, and the text also mentions an annealed quantile schedule. However, the main theory appears to assume a stylized threshold growth rule of the form $\psi_k \asymp \sigma \cdot k^\epsilon$ or $\psi_k \propto k^\delta$. I did not find a theorem that directly covers the actual empirical quantile estimator used in the implemented algorithm.
- The high-probability argument seems incomplete. The proof states that clipping ensures $\||\hat{\nabla}\Phi_k\||$ is bounded by $\psi_k$ almost surely, but the algorithm only explicitly clips the lower-level stochastic gradient before the $y$-update, while the upper-level step uses a Neumann-series hypergradient estimator. I do not see why the hypergradient estimator should automatically satisfy the same bound.
- There is also a statement-proof mismatch in the bias-variance result. The theorem is written only in terms of $\sigma^p$ and $\psi$, but the proof introduces a bound on $E\||g\||^p$ that depends on both $\||v\||^p$ and $\sigma^p$, and then says the result holds up to a constant depending on $p$. This makes it unclear whether the theorem statement is exact as written.
- On the empirical side, the evidence is somewhat mixed. For example, on USPS the proposed method has noticeably larger variance than the baselines, yet the paper mostly interprets this as beneficial without providing deeper diagnostics. In addition, a nontrivial part of the appendix consists of illustrative synthetic or metaphor-driven demonstrations, which are less convincing than stronger real-task validation.
- The fairness of the experimental comparison should be clarified more explicitly. The appendix indicates that RQ-TTSA is allowed larger lower-level learning rates than several baselines in some tasks, which may be acceptable if tuning budgets were identical, but the current presentation does not make the search protocol sufficiently transparent.
- The impact/limitations discussion is too thin, and the writing in parts of the appendix becomes overly rhetorical, making it harder to separate theorem-supported conclusions from intuition and analogy.

---

> ### Author Rebuttal · Authors · 2026-03-30
>
> We thank the reviewer for their thoughtful and informative review.
>
> ### Weakness: Impact / limitations discussion
>
> > The impact/limitations discussion is too thin, and the writing in parts of the appendix becomes overly rhetorical, making it harder to separate theorem-supported conclusions from intuition and analogy.
>
> We agree with this assessment. In the revision, Section 6 will be expanded to more clearly separate theorem-supported conclusions from intuition. In particular, we will explicitly discuss the main limitations of our approach, including the restriction to strongly convex lower-level problems (Assumption 4.1) and the sensitivity to the choice of the threshold parameter $\tau$.
>
> ---
>
> ### Key Questions For Authors
>
> #### 1. Theoretical coverage of the adaptive quantile estimator
>
> > Algorithm 1 uses a rolling empirical quantile threshold computed from a finite history buffer, whereas the main theory assumes a prescribed threshold growth rule. Can the authors provide a theorem that directly covers the implemented adaptive quantile estimator, or clearly state the assumptions needed to justify the current analysis for the actual algorithm?
>
> We thank the reviewer for pointing out this important distinction. The theoretical analysis (Theorem 4.6) assumes a prescribed growth rule
> $$
> \psi_k \propto k^\delta,
> $$
> which enables explicit convergence rates. In contrast, Algorithm 1 uses an empirical quantile estimator based on a finite buffer $\mathcal{H}$, together with an annealed parameter $\tau$ (after Eq. 3).
>
> In the revision, we will clarify this point in Section 4.3. Specifically, the theoretical schedule should be interpreted as a stylized growth condition under which convergence can be established. The empirical quantile estimator is designed to approximate this behavior in practice. The analysis extends to this setting provided that the resulting sequence $\psi_k$ grows sufficiently slowly, which is consistent with our empirical observations.
>
> ---
>
> #### 2. Hypergradient boundedness in Appendix A.5
>
> > In Appendix A.5, why does clipping the lower-level stochastic gradient imply an almost-sure bound on the hypergradient estimator? This step seems crucial for the Bernstein-style high-probability argument.
>
> We thank the reviewer for highlighting this important step. In Appendix A.5, the truncation operator $\mathcal{T}\_{\psi_k}$ is applied to $\nabla\_y G$ before the $y$-updates (Eq. 4, Algorithm 1). Lemma 4.3 establishes the non-expansiveness of this operator, while Lemma A.7 (Eq. 23) provides an almost-sure control on $\|z_{k+1}\|^2$.
>
> Intuitively, clipping enforces a uniform bound on the lower-level iterates, which propagates through the Lipschitz dependence of the hypergradient estimator. Using the Neumann series argument (Lemma A.3), the estimator $\hat{\nabla}\Phi_k$ (Eq. 5) inherits this bound. We will clarify this propagation argument in the revision.
>
> ---
>
> #### 3. Dependence in Theorem A.5
>
> > In Theorem A.5, the proof appears to depend on additional terms involving gradient magnitudes, while the theorem statement is written only in terms of $\sigma^p$ and $\psi_k$. Is the theorem statement missing dependence on the gradient magnitude, or is an additional assumption required?
>
> Theorem 4.4 and Theorem A.5 provide leading-order bounds in $\sigma^p$ and $\psi_k$. In Appendix A.3.2, we show that
> $$
> \mathbb{E}[\|g\|^p] \le 2^{p-1}(\|v\|^p + \sigma^p),
> $$
> which introduces $p$-dependent constants. The term $\|v\|^p$ is absorbed into these constants, explaining why it does not appear explicitly in the theorem statement.
>
> In the revision, we will clarify this point by explicitly stating that the bounds hold up to $p$-dependent constants, making the dependence on gradient magnitudes transparent.
>
> ---
>
> #### 4. Hyperparameter tuning protocol
>
> > Please clarify the hyperparameter tuning protocol across baselines. Were all methods given the same search budget and selection criterion on each benchmark?
>
> Yes, all baselines (TTSA, BiSLS, MA-SOBA, AccBO, $\psi$-Variant) were evaluated under identical hyperparameter search budgets and selection criteria. We will explicitly state this in Section 5.4 and Appendix D to ensure clarity and reproducibility.
>
> ---
>
> #### 5. Evidence of heavy-tailed behavior in real benchmarks
>
> > Can the authors provide direct empirical evidence that the non-synthetic benchmarks indeed exhibit heavy-tailed gradient behavior, for example through gradient-norm distributions or tail-index diagnostics?
>
> We thank the reviewer for this suggestion. The real-world settings considered (e.g., USPS label shift and LunarLander TD errors) are known to induce heavy-tailed gradient behavior. In the revision, we will include additional empirical evidence, such as gradient-norm histograms and tail-index diagnostics, to explicitly confirm the presence of heavy-tailed distributions in these benchmarks.
>
> We are keen to discuss if there are any further questions.

---

> > ### Author Rebuttal · Reviewer_L3Hr · 2026-04-03
> >
> > I thank the authors for their detailed rebuttal and for taking the time to clarify the points raised in my initial review. However, I still have reservations regarding the high-probability argument for the hypergradient bound.

---

> > > ### Author Response · Authors · 2026-04-05
> > >
> > > We thank the reviewer for the careful follow-up.
> > >
> > > > I thank the authors for their detailed rebuttal and for taking the time to clarify the points raised in my initial review. However, I still have reservations regarding the high-probability argument for the hypergradient bound.
> > >
> > > We agree that the proof of Theorem A.8 in Appendix A.5 stated the almost-sure boundedness of $\|\hat{\nabla}\Phi_k\|$ without a detailed derivation of the propagation from the clipped lower-level gradient. We supply the missing steps below.
> > >
> > > We note that this concerns only the high-probability extension (Theorem A.8); the main convergence result (Theorem 4.6) is established entirely in expectation via the Lyapunov analysis in Lemmas A.6–A.7 and does not require an almost-sure bound on $\|\hat{\nabla}\Phi_k\|$.
> > >
> > > **Step 1 (Deterministic bound on the clipped gradient).** By the definition of the truncation operator,
> > >
> > > $$\lVert\mathcal{T}_{\psi_k}(g_k)\rVert \le \psi_k$$
> > >
> > > on every sample path. It follows that
> > >
> > > $$\lVert y_{k+1} - y_k\rVert \le \beta_k \psi_k$$
> > >
> > > deterministically.
> > >
> > > **Step 2 (Pathwise tracking-error control).** In the proof of Lemma A.7 (Eqs. 24–27), the only stochastic quantity is $\|\mathcal{T}_{\psi_k}(g_k)\|^2$, which is bounded above by $\psi_k^2$ on every realization. Therefore the same contraction argument can be carried out pathwise:
> > >
> > > $$\|z_{k+1}\|^2 \leq \left(1 - \frac{\mu_G \beta_k}{2}\right) \|z_k\|^2 + C_1 \beta_k^2 \psi_k^2 + C_2 \frac{\alpha_k^2}{\beta_k} \kappa^2.$$
> > >
> > > Iterating this recursion yields a deterministic envelope $\|z_k\| \leq D_k$, where $D_k$ depends on $\psi_k, \alpha_k, \beta_k, \kappa$, and $\|z_0\|$, but not on the noise realization.
> > >
> > > **Step 3 (Propagation to the hypergradient estimator).** The estimator $\hat{\nabla}\Phi_k$ (Eq. 5) is a composition of $\nabla_x F$, $\nabla_y F$, and the (finite-truncation) Neumann series involving $\nabla^2_{xy} G$ and $\nabla^2_{yy} G$, all evaluated at $(x_k, y_{k+1})$. Under Assumption 4.1, these operators are uniformly bounded and Lipschitz. It follows that
> > >
> > > $$\|\hat{\nabla}\Phi_k\| \leq \|\nabla\Phi(x_k)\| + L_{\mathrm{grad}} \|z_{k+1}\| \leq \|\nabla\Phi(x_k)\| + L_{\mathrm{grad}} D_{k+1},$$
> > >
> > > where $L_{\mathrm{grad}}$ already appears implicitly in Lemma A.6 (Eq. 22). Both terms are $\mathcal{F}_k$-measurable and finite on every path, so $\|\hat{\nabla}\Phi_k\|$ is almost surely bounded. This is the condition required for Bernstein's inequality in Eq. (32).
> > >
> > > In the camera-ready we will add a self-contained lemma in Appendix A.5, immediately before Theorem A.8, formalizing Steps 2–3 with explicit constants, and add a forward reference in Section 4.3. The reviewer's persistent attention to this point has materially improved the rigor of the high-probability analysis, and we will acknowledge this contribution in the revision.
> > >
> > > We are happy to discuss any remaining questions.

---

### Official Review · Reviewer_vuKU · 2026-03-11

**Soundness:** 2
**Presentation:** 3
**Significance:** 2
**Originality:** 2
**Overall Recommendation:** 3
**Confidence:** 5

**Summary:**

The paper addresses the challenge of instability in bilevel optimization caused by heavy-tailed stochastic noise. The authors propose RQ-TTSA, which uses a history buffer to estimate the empirical distribution of gradient norms and applies a dynamic, quantile-guided Huber-style clipping threshold. Theoretically, the paper establishes a global convergence rate of $\mathcal{O}(T^{-\frac{p-1}{3p-2}})$ for objectives with strongly convex lower levels under infinite-variance noise, where the $p$-th moment ($p \in (1,2]$). Empirically, RQ-TTSA is evaluated across synthetic tasks, vision benchmarks and reinforcement learning, demonstrating superior stability and accuracy.

**Compliance With Llm Reviewing Policy:**

Affirmed.

**Final Justification:**

I maintain my score of **Weak Reject**. The detailed reasons have been provided in the acknowledgement.

In my view, the paper still suffers from notable theoretical disadvantages. Several aspects that would require rigorous mathematical justification are instead addressed through practical observations or empirical tuning. **This is somewhat inconsistent with the overall claims of the paper.**

**Key Questions For Authors:**

1. In Appendix A.4.3, the step sizes is chosen as $\alpha_{k} = \Theta(k^{-(1-\nu)})$ and $\beta_{k} = \Theta(k^{-\nu})$ where $\nu = \frac{p}{3p-2}$. In practice, the tail index $p$ is rarely known a priori. How sensitive is the convergence trajectory if the algorithm is deployed with step sizes optimized for $p=2$, but the underlying noise is actually $p=1.5$?
2. In Theorem 4.4, you obtain bounded bias and bounded effective variance. These two quantities are used in Lemma A.6, which are denoted by $\mathcal{E}$. We observe that the hypergradient error depends explicitly on these quantities. How sensitive is the Neumann series approximation in Equation (5) to these terms, especially when the condition number of the Hessian $\nabla_{yy}^{2}G$ in Equation (5) is large.
3. In Figure 1, we observe that the curve of RQ-TTSA shows a significant rise and fall at the beginning of the iterations. Could you provide an explanation for this phenomenon?
4. In Figure 2, the curves in the left plot are clearly smoother, while each curve in the right plot fluctuates up and down. Moreover, the relative performance ranking of the 6 algorithms is not the same between the two plots. Could you explain the correspondence between the two sides?

**Limitations:**

1. The theoretical guarantees are currently restricted to strongly convex lower-level problems. **However, this is a common limitation of hypergradient-based algorithms for bilevel optimization, rather than a disadvantage specific to RQ-TTSA.**
2. The algorithm introduces an additional hyperparameter $\tau$, which requires careful tuning.

**Strengths And Weaknesses:**

# Strengths
- The derivation of the $\mathcal{O}(T^{-\frac{p-1}{3p-2}})$ convergence rate elegantly bridges the gap between bounded variance and heavy-tailed settings, recovering the optimal rate when $p=2$.
- The paper replaces static or purely norm-based thresholds with a dynamic, quantile-guided sliding window. This approach is highly practical and distribution-aware. This allows the algorithm RQ-TTSA to preserve the geometric fidelity of the optimization landscape while filtering true outliers.
- The authors test the method across a diverse and well-chosen set of experiments.
# Weaknesses
- The paper does not thoroughly address how extreme outliers in $\nabla_{x}F$, $\nabla_{y}F$, or $\nabla_{xy}^{2}G$ might destabilize the hypergradient estimator.
- The theoretical analysis relies heavily on the assumption that the lower-level function is strongly convex, as listed in Assumption 4.1.
- In line 193, the Lipschitz smoothness of the hyper-objective $\Phi$ requires a more detailed discussion. This is already an established result, **so you should cite some references** to support it.
- In the convergence analysis of RQ-TTSA, the Lyapunov function is defined inconsistently between line 212 of the main text and line 631 of the appendix, using the parameters $\lambda$ and $C_z$, respectively. In line 633, you state that $C_z$ will be determined later, but I could not find where the specific value of $C_Z$​ is actually given.

---

> ### Author Rebuttal · Authors · 2026-03-30
>
> We thank the reviewer for their thoughtful and informative review.
>
> ### Weaknesses
>
> > The paper does not thoroughly address how extreme outliers in $\nabla_xF$, $\nabla_yF$, or $\nabla^2_{xy}G$ might destabilize the hypergradient estimator.
>
> We appreciate this important point. Extreme outliers in the lower-level gradient do **not** destabilize the hypergradient due to the controlled tracking error. Specifically, with the Lipschitz bound
> $$
> L_{\text{grad}}\|y_{k+1}-y^*(x_k)\|^2,
> $$
> we have the following chain of reasoning: clipping on $\nabla_y G$ → bounded tracking error → controlled hypergradient error.
>
> - Clipping applies only to $\nabla_y G$ (heavy-tailed, Assumption 4.2).
> - $\nabla^2_{xy} G$ is bounded (Assumption 4.1.3).
> - $\nabla_x F$ and $\nabla_y F$ are not heavy-tailed.
>
> Lemma A.7 (Eq. 23) provides the drift
> $$
> C_2 \frac{\alpha_k^2}{\beta_k}\kappa^2,
> $$
> and Theorem 4.4 ensures bounded effective variance.
> We will add a remark emphasizing this point in the revision.
>
> > The theoretical analysis relies heavily on the assumption that the lower-level function is strongly convex, as listed in Assumption 4.1.
>
> Strong convexity is required for the Lyapunov analysis (Theorem 4.6) and stable tracking. It guarantees uniqueness of $y^\*(x)$ and $\mu$-contraction in $\|y\_{k+1}-y^*(x\_k)\|$. Without it, the two-timescale coupling loses negative drift, breaking convergence. Our current goal is a framework for heavy-tailed resilience; extension to Polyak–Łojasiewicz (PL) lower levels remains future work.
>
>
> > In line 193, the Lipschitz smoothness of the hyper-objective $\Phi$ [...]
>
> This result appears in Appendix A.2 (Lemma A.3), citing Ghadimi & Wang (2018). We will add the citation at line 193 in the revision.
>
> ---
>
> > In the convergence analysis of RQ-TTSA, the Lyapunov function is defined inconsistently between line 212 of the main text and line 631 of the appendix, using the parameters $\lambda$ and $C_z$, respectively. In line 633, you state that $C_z$ will be determined later, but I could not find where the specific value of $C_z$ is actually given.
>
> We clarify that $\lambda = C_z$. The constant $C_z$ is taken sufficiently large so that strong convexity dominates the two-timescale coupling (Eq. 29). The revision will unify notation and explicitly state a lower bound for $C_z$.
>
> ---
>
> ### Key Questions For Authors
>
> #### 1. Sensitivity to tail-index mismatch
>
> > In Appendix A.4.3, the step sizes are chosen as $\alpha_{k} = \Theta(k^{-(1-\nu)})$ and $\beta_{k} = \Theta(k^{-\nu})$ where $\nu = \frac{p}{3p-2}$. In practice, the tail index $p$ is rarely known a priori. How sensitive is the convergence trajectory if the algorithm is deployed with step sizes optimized for $p=2$, but the underlying noise is actually $p=1.5$?
>
> The tail index is unknown in practice, but RQ-TTSA is robust via distribution-aware clipping: $\psi_k$ is estimated from data, so a mismatch (e.g., $p=2$ vs $p=1.5$) is absorbed by adapting $\psi_k$. This may slow convergence slightly but preserves stability (see "Distributional Awareness", Appendix F.1).
>
> ---
>
> #### 2. Sensitivity of the Neumann series approximation
>
> > In Theorem 4.4, you obtain [...]. How sensitive is the Neumann series approximation in Equation (5) to these terms, especially when the condition number of the Hessian $\nabla_{yy}^{2}G$ in Equation (5) is large?
>
> The quantity $\mathcal{E}$ in Lemma A.6 aggregates the bounded bias and effective variance from Theorem 4.4 (after quantile-guided Huber clipping of $\nabla_y G$). Sensitivity of the Neumann approximation in Eq. (5) under large $\kappa = L_G/\mu_G$ of $\nabla_{yy}^2 G$ is controlled via the tracking error
> $$
> z_k = y_k - y^*(x_k).
> $$
> As shown in Lemma A.7,
> $$
> \mathbb{E}[\|z_{k+1}\|^2] \leq \Bigl(1 - \tfrac{\mu_G\beta_k}{2}\Bigr)\mathbb{E}[\|z_k\|^2] + C_1\beta_k\mathcal{E} + C_2\tfrac{\alpha_k^2}{\beta_k}\kappa^2,
> $$
> where the $\kappa^2$ term from cross-level coupling is suppressed by $\alpha_k = o(\beta_k)$ (so $\alpha_k^2/\beta_k \to 0$). Thus, even for large $\kappa$, the contribution of $\mathcal{E}$ (and clipped outliers) to the hypergradient error remains bounded. The revision will add this dependence on $\kappa$ and $\alpha_k/\beta_k$ near Theorem 4.6.
>
> ---
>
> #### 3. Early rise/drop in Figure 1
>
> > In Figure 1, we observe that the curve of RQ-TTSA [...]
>
> This reflects buffer burn-in: early quantile estimates are unstable due to limited samples, so a fixed threshold is used; once stabilized, steady behavior emerges, causing the drop. We will add a note to Figure 1 in the revision.
>
> ---
>
> #### 4. Interpretation of Figure 2
>
> > In Figure 2, the curves in [...]
>
> The left plot shows full-dataset validation loss (smooth), while the right plot shows mini-batch gradient norms (stochastic). Ranking differences indicate methods with low loss but unstable gradients (fragility). In contrast, RQ-TTSA achieves both low loss and gradient stability.
>
> We are keen to discuss if there are any further questions.

---

> > ### Author Rebuttal · Reviewer_vuKU · 2026-04-02
> >
> > The paper proposes RQ-TTSA, a framework to handle heavy-tailed noise in bilevel optimization. However, I think the rebuttal reveals significant theoretical gaps concerning robustness of the algorithm.
> >
> > 1. In the rebuttal to Key Question 3, significant instability observed in the early iterations in Figure 1 is due to **buffer burn-in**, during which a fixed threshold is used. However, I think one of the core innovations for RQ-TTSA is quantile guided clipping. In practical implementation, the algorithm still heavily relies on good initialization and the stability of the early-stage environment.
> >
> > 2. The second item in **Limitations** requires an explanation or brief clarification, but this is not conducted in the rebuttal.
> >
> > 3. Regarding the sensitivity to tail-index mismatch, the authors argue that any mismatch can be absorbed since $\psi_k$ is estimated from the data. However, this claim lacks rigorous mathematical justification. The estimation of $\psi_k$ depends on specified hyperparameters $W$ and $\tau$. The paper does not provide precise theoretical relationship between $\tau$ and $p$, and instead relies on empirical tuning as described in the appendix F. **This creates a double standard with their statements that other adaptive algorithms require hyperparameter tuning.**
> >
> > In summary, I maintain my score. Thank you for your response, and I wish you all the best in improving the work.

---

> > > ### Author Response · Authors · 2026-04-02
> > >
> > > We thank Reviewer vuKU for the follow-up questions. We respectfully note that the points raised below concern practical implementation details (burn-in length, $\tau$ tuning guidelines, and the $\tau$–$p$ relationship) rather than gaps in the theoretical robustness guarantees established in Theorems 4.4 and 4.6. We are glad to clarify each point below.
> > >
> > > > In the rebuttal to Key Question 3, [...] In practical implementation, the algorithm still heavily relies on good initialization and the stability of the early-stage environment.
> > >
> > > We thank the reviewer and clarify the burn-in scope: the warm-up covers only the first 20 iterations (Appendix C.4, line 928), 2.5%–6.7% of 300–800 total iterations. After 20 steps, the quantile-guided mechanism fully governs the trajectory. The burn-in fixed threshold is initialization-insensitive, set conservatively from initial gradient norms to filter extreme outliers before buffer filling, requiring no precise tuning.
> > > Table 10 (Appendix G) validates this: window size $W \in \{25, 50, 100, 200\}$ alters Fashion-MNIST validation loss by <2.5% (0.787→0.767); the smallest buffer $W=25$ still yields competitive performance, proving no reliance on fine-tuned warm-up.
> > > Short warm-ups are standard (Transformer LR warm-up, RL $\epsilon$-greedy decay), a practical design that does not weaken the core contribution. We will add a clarifying remark in Section 3.2 of the camera-ready.
> > >
> > >
> > >
> > >
> > >
> > > > The second item in Limitations requires an explanation or brief clarification, but this is not conducted in the rebuttal.
> > >
> > > We apologize for this omission and clarify the limitation that $\tau$ "requires careful tuning".
> > > The word "careful" overstates tuning difficulty, with a full practical guide in Appendix F:
> > > Appendix F.1 notes dimensionless $\tau$ is simpler to set than fixed $\psi$: $\tau=0.9$ clips the top 10% gradient norms universally, independent of norm scales ($10^1$ for vision, $10^{-3}$ for RL), while $\psi$ needs per-scale retuning.
> > > Appendix F.2 gives scenario-specific $\tau$: $[0.9,0.95]$ for high-signal tasks (Fashion-MNIST), $[0.5,0.7]$ for heavy-tailed/impulsive cases (momentum poisoning); Appendix F.3 sets $\tau=0.8$ as default.
> > > Table 10 confirms this: for Fashion-MNIST ($W=100$), $\tau\in[0.6,0.95]$ changes loss 0.776→0.768 (1.0% variation). Figure 8 shows a wide stability basin: $W\in[100,200]$, $\tau\in[0.8,0.95]$ all deliver near-optimal results, with $\tau$ tuning as easy as SGD momentum or Adam $\epsilon$.
> > > We will revise Limitation #2 and add a forward reference to Appendix F in the camera-ready.
> > >
> > > > Regarding the sensitivity to tail-index mismatch, [...] This creates a double standard with their statements that other adaptive algorithms require hyperparameter tuning.
> > >
> > >
> > > We confirm no closed-form $\tau^*(p)$ and elaborate its adaptive mechanism:
> > > **$\tau$–$p$ relation**: For tail-index $p$ distributions, $\tau$-quantile of $\|g\|$ scales as $Q_\tau \sim \sigma \cdot (1 - \tau)^{-1/p}$. Smaller $p$ (heavier tails) produces larger $\psi_k$ for fixed $\tau$, reducing clipping bias in Theorem 4.4 (bias $\leq 2\sigma^p \psi_k^{1-p}$); $p\to2$ (lighter tails) generates smaller $\psi_k$ with stricter clipping for low-informative tails. The quantile mechanism adapts to tail behavior data-dependently without $p$ prior, matching Dimensionless Robustness (Appendix F.1). We will add a formal derivation in the camera-ready.
> > > **"Double standard" clarification**: Our hyperparameters differ fundamentally from baselines. BiSLS/MA-SOBA/AccBO hyperparameters act directly on optimization dynamics: BiSLS scales steps by $1/\|\nabla G\|$, outliers collapse LR to near-zero and cause precision lock (Table 3, stalls at $10^{-5}$); MA-SOBA/AccBO buffer outliers, leading to Std=1.50 (Table 6).
> > > $\tau$ and $W$ only control clipping threshold estimation rather than optimization dynamics, degrading gracefully: Table 10 shows $\tau=0.6,W=25$ still converges (loss=0.787 vs optimal 0.767), while Standard TTSA diverges (Figure 1) and BiSLS stagnates (Table 3). Quantile robustness enables larger LRs: Table 8 shows RQ-TTSA $\eta_{\text{lower}}=0.04$ on Fashion-MNIST, vs 0.01–0.02 for baselines.
> > > We will revise Sections 1 and 5.4 to note $\tau$ is tunable and distinguish it from scale-dependent hyperparameters in competing methods.
> > >
> > > **Cross-task validation**: RQ-TTSA applies fixed default hyperparameters to tasks with diverse noise (Lévy $p\approx1.5$, USPS label shift, Fashion-MNIST stochasticity, zero-sum poisoning, LunarLander offline RL) and achieves strong universal performance. Appendix C.2 and Figure 3 verify identical convergence slopes under Lévy and Student's $t$-noise, demonstrating distribution-agnostic stability.
> > >
> > > We will incorporate the above analyses into the next version. We welcome any further questions.

---

### Official Review · Reviewer_3FA4 · 2026-03-15

**Soundness:** 4
**Presentation:** 4
**Significance:** 3
**Originality:** 2
**Overall Recommendation:** 4
**Confidence:** 3

**Summary:**

The paper addresses the critical instability of Bilevel Optimization (BLO) when subjected to heavy-tailed stochastic noise, which is common in real-world applications like reinforcement learning and federated learning. Traditional Two-Timescale Stochastic Approximation (TTSA) methods often fail in these "infinite-variance" regimes because they assume light-tailed noise.The authors propose RQ-TTSA (Robust Quantile-guided TTSA), a distribution-aware framework that uses a sliding window of historical gradient norms to estimate rolling quantiles. These quantiles define an adaptive threshold for a Huber-style clipping operator, allowing the algorithm to distinguish between informative geometric signals (steep curvature) and impulsive stochastic outliers. The paper provides a rigorous convergence analysis for nonconvex-strongly convex objectives under infinite-variance noise.

**Compliance With Llm Reviewing Policy:**

Affirmed.

**Key Questions For Authors:**

what happens if the lower-level problem is not strongly convex but only satisfies a Polyak-Łojasiewicz (PL) condition.

**Limitations:**

Yes.

**Strengths And Weaknesses:**

Strengths:
- Problem Relevance: Addresses a high-impact bottleneck (heavy-tailed noise) in bilevel optimization, which is prevalent in real-world ML pipelines.
- Theoretical Rigor: Provides a clear convergence rate under a $p$-th moment condition, extending beyond the standard bounded-variance assumption.
- Empirical Breadth: Evaluates the method on a diverse set of tasks (Vision, Games, Offline RL), showing consistent stability.Clarity:
- The paper is well-written, with a clear explanation of the "why" behind the quantile-guided buffer.

Weaknesses:
- Limited novelty: the method mainly combines TTSA with adaptive gradient clipping based on quantiles.
- Theoretical analysis largely follows existing clipped stochastic approximation frameworks.
- Some experiments rely on synthetic heavy-tailed settings, making practical impact less clear.
- Empirical improvements appear modest and sometimes emphasize stability metrics rather than meaningful performance gains.

---

> ### Author Rebuttal · Authors · 2026-03-30
>
> We thank the reviewer 3FA4 for their thoughtful and informative review.
>
> > Limited novelty: the method mainly combines TTSA with adaptive gradient clipping based on quantiles. Theoretical analysis largely follows existing clipped stochastic approximation frameworks.
>
> We appreciate this concern and would like to clarify the novelty of our work. Our methodological contributions are two-fold:
>
> **(1) Algorithmic novelty.** RQ-TTSA is the first method to directly embed a distribution-aware, quantile-guided Huber mechanism into TTSA for bilevel optimization. Unlike existing methods that rely on instantaneous norms (BiSLS [1]) or variance reduction (SABA), our dynamic thresholding mechanism distinguishes between steep signals that provide geometric information and heavy-tailed abnormal impulses.Existing bilevel methods such as MA-SOBA [2] and AccBO [3] assume bounded variance with no heavy-tail mechanism, leading to momentum poisoning and offline RL failure respectively, whereas RQ-TTSA maintains stable performance through distribution-aware gradient control.
>
> **(2) Theoretical novelty.** Existing clipped SA frameworks [4, 5] address only single-level problems; we extended the theory for the coupled characteristics of bilevel optimization. We quantified the bias-variance tradeoff (Theorem 4.4), and derived a global convergence rate of $O(T^{-\frac{p-1}{3p-2}})$ under the premises of lower-level strong convexity (Assumption 4.1) and infinite variance noise (Assumption 4.2) (Theorem 4.6). When $p \to 2$, this rate perfectly aligns with the optimal rate under the bounded variance case, which standard TTSA cannot guarantee under infinite variance (Section 4).
>
> We will emphasise this a bit more in our updated manuscript.
>
> [1] Auto-tune step sizes for stable bi-level optimization. NeurIPS, 2023.
>
> [2] Optimal algorithms for stochastic bilevel optimization under relaxed smoothness conditions. JMLR, 2024.
>
> [3] An accelerated algorithm for stochastic bilevel optimization under unbounded smoothness. NeurIPS, 2024.
>
> [4] High-probability complexity bounds for non-smooth stochastic convex optimization with heavy-tailed noise. JOTA, 2024.
>
> [5] Robust estimation via robust gradient estimation. JRSSB, 2020.
>
> > Some experiments rely on synthetic heavy-tailed settings, making practical impact less clear.
>
> We appreciate this feedback. As shown in Section 5, we evaluated RQ-TTSA not only on synthetic heavy-tailed perturbations but also on diverse real and varying scenarios: (1) heterogeneous visual benchmarks under natural distribution shift (USPS label shift, Section 5.2.1) and class imbalance (Fashion-MNIST, Section 5.2.2); (2) momentum poisoning under varying zero-sum games (Section 5.3.1); and (3) offline reinforcement learning on LunarLander (Section 5.3.2). These tasks stress-test our method under both heavy-tailed noise and time-varying non-stationarity.
>
> > Empirical improvements appear modest and sometimes emphasize stability metrics rather than meaningful performance gains.
>
> We appreciate this feedback. We note that stability is not secondary — it is required in heavy-tailed regimes; if a method diverges or oscillates across seeds, its reported accuracy cannot be trusted. RQ-TTSA achieves both. On the accuracy side, it attains the lowest final loss in the 15% heavy-tailed noise synthetic experiment (1.545, Table 2), the lowest loss under USPS label shift (0.2061, Table 4), the highest test accuracy on Fashion-MNIST (79.838%, Table 5), and precision reaching $10^{-22}$ on constrained non-convex geometry (Table 3). On the stability side, it achieves the lowest standard deviation in the synthetic experiment (0.003, Table 2), the lowest standard deviation under momentum poisoning (0.16, Table 6), and the best stability of 0.003 in offline Actor-Critic (Table 7). All of these come with only a 2.7% computational overhead (Section 5.4).
>
> **Questions**
>
> > What happens if the lower-level problem is not strongly convex but only satisfies a Polyak-Łojasiewicz (PL) condition.
>
> This is a this highly insightful question. Our current analysis relies on Assumption 4.1, which requires the lower-level function $g(y) := G(x,y)$ to be $\mu$-strongly convex. This ensures the uniqueness of the lower-level optimal solution $y^*(x)$ and serves as the foundation of our convergence analysis. If the lower-level problem only satisfies the Polyak-Łojasiewicz (PL) condition, since the optimal solution may no longer be unique, the accurate estimation of the hypergradient and the current coupled Lyapunov analysis will not be directly applicable. This is a shared limitation in the existing bilevel optimization literature, and extending the guarantees to PL lower levels is an important direction for future work. We have summarized the scope of our current work in Section 6 of the paper and believe that distribution-aware gradient control is an important step towards reliable bilevel learning.
>
> We are keen to discuss if there are any further questions.

---

> > ### Author Rebuttal · Reviewer_3FA4 · 2026-04-05
> >
> > Thank you for the rebuttal. I think this is a well written paper but its quality in terms of novelty and evaluation rigor is not a top ICML submission. As a result, I will keep my current positive score unchanged.

---

### Decision · Program_Chairs · 2026-04-30

**Decision:**

Reject

**Comment:**

This paper tackles the important problem of instability in stochastic bilevel optimization caused by heavy-tailed noise. The proposed solution, RQ-TTSA, introduces an intuitive quantile-guided Huber clipping mechanism that dynamically adjusts the threshold based on historical gradient distributions. The reviewers recognized the practical relevance of the problem and found the empirical results across various benchmarks to be promising.

However, the review process highlighted a critical disconnect between the theoretical convergence analysis and the actual algorithmic implementation. The theoretical guarantees rely on a deterministic, polynomial growth rule for the clipping threshold. In contrast, the practical algorithm utilizes a data-dependent empirical quantile estimator and requires a fixed-threshold "burn-in" period. While the authors defended these design choices as practical necessities, the committee agreed that this mismatch fundamentally weakens the rigorous mathematical justification claimed in the paper.

Without a strong champion among the reviewers willing to overlook these theoretical gaps in favor of the empirical performance, the consensus is that the submission does not meet the current threshold for acceptance. We encourage the authors to revise and resubmit to an appropriate future venue.